# Correlation of Electrophysiological and Fluorescence-Based Measurements of Modulator Efficacy in Nasal Epithelial Cultures Derived from People with Cystic Fibrosis

**DOI:** 10.3390/cells12081174

**Published:** 2023-04-17

**Authors:** Tarini N. A. Gunawardena, Zoltán Bozóky, Claire Bartlett, Hong Ouyang, Paul D. W. Eckford, Theo J. Moraes, Felix Ratjen, Tanja Gonska, Christine E. Bear

**Affiliations:** 1Program of Molecular Medicine, The Hospital for Sick Children, Toronto, ON M5G 1X8, Canada; 2Program of Translational Medicine, The Hospital for Sick Children, Toronto, ON M5G 1X8, Canada; 3Providence Health Care, Vancouver, BC V6Z 1Y6, Canada; 4Department of Paediatrics, Division of Respiratory Medicine, The Hospital for Sick Children, Toronto, ON M5G 1X8, Canada; 5Department of Paediatrics, Division of Gastroenterology, Hepatology and Nutrition, University of Toronto, Toronto, ON M5S 1A8, Canada; 6Department of Physiology, University of Toronto, Toronto, ON M5S 1A8, Canada; 7Department of Biochemistry, University of Toronto, Toronto, ON M5S 1A8, Canada

**Keywords:** cystic fibrosis, nasal epithelial cultures, CFTR modulators, theratyping

## Abstract

It has been suggested that in vitro studies of the rescue effect of CFTR modulator drugs in nasal epithelial cultures derived from people with cystic fibrosis have the potential to predict clinical responses to the same drugs. Hence, there is an interest in evaluating different methods for measuring in vitro modulator responses in patient-derived nasal cultures. Commonly, the functional response to CFTR modulator combinations in these cultures is assessed by bioelectric measurements, using the Ussing chamber. While this method is highly informative, it is time-consuming. A fluorescence-based, multi-transwell method for assaying regulated apical chloride conductance (Fl-ACC) promises to provide a complementary approach to theratyping in patient-derived nasal cultures. In the present work, we compared Ussing chamber measurements and fluorescence-based measurements of CFTR-mediated apical conductance in matching, fully differentiated nasal cultures derived from CF patients, homozygous for F508del (*n* = 31) or W1282X (*n* = 3), or heterozygous for Class III mutations G551D or G178R (*n* = 5). These cultures were obtained through a bioresource called the Cystic Fibrosis Canada-Sick Kids Program in Individual CF Therapy (CFIT). We found that the Fl-ACC method was effective in detecting positive responses to interventions for all genotypes. There was a correlation between patient-specific drug responses measured in cultures harbouring F508del, as measured using the Ussing chamber technique and the fluorescence-based assay (Fl-ACC). Finally, the fluorescence-based assay has the potential for greater sensitivity for detecting responses to pharmacological rescue strategies targeting W1282X.

## 1. Introduction

The aim of CFTR modulator theratyping using patient-derived tissue cultures is to support precision medicine for people harbouring rare, disease-causing mutations in CFTR [1,2]. The overarching goal is to provide patient-specific, in vitro evidence of the therapeutic efficacy of modulators for clinical decision-making or regulatory approval. Ultimately, acceptance by regulatory bodies of the results of drug testing on patient-derived tissue cultures will depend on the quality of the in vitro results and their correlation with patient-specific clinical outcomes. 

A plate-reader-based assay of the functional response of CF-patient-derived, 3D intestinal organoid cultures to CFTR modulators was originally developed by the research group of Beekman and colleagues [3,4]. Forskolin-induced swelling of intestinal organoids reports CFTR-regulated fluid transport, and this phenotype has been used by the Beekman group and multiple labs to rank the relative efficacy of panels of compounds on organoids generated from people with different loss-of-function mutations. Such medium–high-throughput assays of patient tissue enable simultaneous ranking of multiple modulators using instrumentation that is commonly available in academic labs. However, access to this type of patient tissue can be limited in certain clinical centres.

Nasal epithelial cells can be accessed readily from consenting patient donors. Robust methods exist for expansion of nasal basal cells and their differentiation as 2D cultures at the air–liquid interface on transwell filters [1,5,6,7,8]. The functional rescue of CFTR in these cultures is measured by electrophysiological methods using the Ussing chamber apparatus. These studies require specialized equipment and expertise. Efforts to develop robust plate-reader assays of CFTR-mediated fluid transport in patient-specific 3D nasal organoids are showing promise [9]. 

Previously, we published describing description of the modification of a fluorescence-based assay of CFTR channel function [10], for the study of 2D patient-derived nasal epithelial cultures differentiated on 96-transwell plates. In this initial study, we showed that this fluorescence-based assay of CFTR-mediated apical chloride conductance was effective in reporting a positive response to treatment with lumacaftor and ivacaftor in 2D nasal cultures generated on 96-transwell plates. These studies included data from a relatively small number of individuals with CF (*n* = 8).

In the present study, we conducted paired studies, comparing the responses by differentiated nasal cultures from 39 individuals (sampling multiple genotypes) using either the fluorescence-based assay of CFTR function or the Ussing chamber assay. Our goal was to determine whether the functional responses to modulators measured using these two methods were correlated with one another. 

## 2. Materials and Methods

### 2.1. Nasal Cell Culture

Human nasal epithelial cell (HNEC) samples were generated from 39 CF individuals by carrying out nasal brushings from the nasal inferior turbinate, as previously described [11]. The HNECs were expanded until passage 1 in PneumaCult™ Ex media (STEMCELL Technologies, Vancouver, BC, Canada) containing the antibiotic cocktail penicillin (100 units/mL), streptomycin (100 µg/mL), amphotericin (0.25 µg/mL), tobramycin (80 µg/mL), vancomycin (16 µg/mL), metronidazole (32 µg/mL), meropenem (8 µg/mL), septra (trimethoprim/sulfamethoxazole) (16/80 µg/mL), and colistimethate (6 µg/mL), supplemented with 5 μM rho kinase inhibitor Y27632 (Selleck Chemicals, Houston, TX, USA). The HNECs were stocked at passage 1 until further use. 

Upon experimentation, the HNECs were expanded further to passage 2 in PneumaCult™ Ex Plus media (STEMCELL Technologies) containing the aforementioned antibiotics. Following the expansion, passage 3 cells were seeded in collagen IV (Sigma-Aldrich, St. Louis, MI, USA)-coated 24-transwell inserts (6.5 mm diameter, 0.4 μm pore size, Corning Inc. New York, NY, USA) at a seeding density of 1 × 10^5^ cells per well for Ussing chamber studies, and in collagen IV (Sigma-Aldrich)-coated 96-transwell inserts (4.26 mm diameter, 0.4 μm pore size, Corning Inc.) at a seeding density of 8 × 10^4^ cells per well for plate-reader-based fluorescent membrane potential (FMP) studies. 

The cells seeded in the transwell inserts were maintained in PneumaCult™ Ex Plus media for 14 days (F508del, G551D and G178R genotypes) or 28 days for cells from people harbouring W1282X. Cells from people with this nonsense mutation tended to require more time to differentiate and acquire a basal resistance of 200 Ω.cm^2^. Upon complete confluency, the differentiation medium PneumaCult™ ALI (STEMCELL Technologies) supplemented with penicillin (100 units/mL) and streptomycin (100 µg/mL) was added to the basolateral region of the cells, and the cultures were maintained at an air–liquid interface (ALI) until the functional studies were carried out at ALI day 14 for the F508del/F508del and Class III samples, while the functional studies for W1282X/W1282X were carried out at ALI day 28. 

#### Drug Treatment Conditions

Chronic drug treatments were carried out basolaterally 48 h before the functional studies. The drug treatment conditions are shown in Table 1 (all drugs were added to 750 µL of PneumaCult™ ALI media to reach the final concentration of each drug, as stated in Table 1. We did not exceed 0.1% DMSO by volume for any of the additions. 

### 2.2. Fluorescence-Based Apical Chloride Conductance (Fl-ACC) Assay

The HNECs were treated for 48 h before carrying out the plate-reader-based Fl-ACC assay, as shown in Table 1 (technical replicates of *n* = 3 for each drug condition). A chloride gradient was maintained by the addition of 200 µL of Hanks’ buffered solution to the basolateral region of the 96-transwell plate (4.26 mm diameter, 0.4 μm pore size, Corning) and the addition of 95 µL of 0.5 mg/mL of the membrane potential dye dissolved in the chloride-free NMDG buffer (150 mM NMDG-Gluconate, 3 mM KCl, 10 mM HEPES, pH 7.35, osmolarity 300 mOsm). The Fl-ACC assay involved the measurement of membrane potential using a fluorescent, voltage-sensitive small molecule—the FLIPR^®^ dye (R8034, Molecular Devices). Following the addition of the FLIPR^®^ dye (excitation; 510–545 nm, emission; 565–625 nm), the cells were incubated in the dark at 37 °C, 5% CO_2_ for 30 min and then transferred to the SpectraMax i3X Multi-Mode Assay Microplate reader or the FLIPR Tetra^®^ instrument (Molecular Devices, San Jose, CA, USA). The peak fluorescence in each well (SpectraMax i3X) or well scans of fluorescence-based membrane potential (FMP) were obtained at baseline, or at CFTR channel activation or inhibition. The CFTR channels were activated by 10 µM Forskolin (FSK) and 1 µM VX-770 or inhibited by the addition of 10 µM CFTRinh172. Fluid transfer in the FLIPR Tetra was carried out as a single aspirate–single dispense, with 5 µL of drug being added during each drug addition. 

The FLIPR Tetra uses an electron-multiplying CCD (EMCCD) sensor for enhanced sensitivity of fluorescence imaging. In the experiments using the FLIPR Tetra, the time courses for CFTR channel activation and inhibition were recorded as multiframe images, where each pixel in a well scan was turned into an individual data trace. The data generated were exported and analysed using the analysis methods described below. 

Analysis of fluorescence membrane potential (FMP) data: The analysis of the FMP data obtained using the SpectraMax i3X (shown in Appendix A) was as previously described [12]. The experimental raw output of the FLIPR Tetra instrument was a single multiframe TIFF file with 105 image frames. The resolution of each frame was 512 by 341 pixels. Since the physical plate’s dimensions were 127.76 by 85.47 mm, it resulted in a 0.25 by 0.25 mm^2^ pixel size. Out of the 105 frames, 5 were dedicated to measuring the baseline (30 s read intervals), 70 to measuring stimulation (15 s read intervals), and 30 to measuring the inhibition (30 s read intervals) intensities. The fluid transfer was carried out as a single aspirate–single dispense, with 5 µL of drug added during each drug addition. To eliminate the intrinsic fluorescence of the plate’s plastic, the background intensity was subtracted from the entire plate. The background intensity was calculated as the maximal intensity of the first frame, excluding areas where fluorescent-dye-loaded cells were present (wells). Throughout the experiment, the fluorescence intensity change at a given location (trace) of each pixel was reported independently. These traces were then normalized using the last point of the baseline to eliminate any absolute intensity differences due to dye loading, cell number differences, or other artifacts. Responses to stimulation and inhibition were calculated as the maximal change during the corresponding experimental segment. Then, the data from pixels belonging to the same transwell were averaged, and transwells corresponding to the same experimental conditions were aggregated and compared. The closest two values of the three replicates were used for subsequent statistical analysis.

### 2.3. Ussing Chamber Studies

The HNECs were treated 48 h before carrying out the Ussing chamber studies, as shown in Table 1 (*n* = 1 for each drug treatment condition). The 6.5 mm transwells were mounted in the Ussing chamber (Physiological Instruments, San Diego, CA, USA), and the experiments were carried out in an open-circuit mode maintaining symmetrical chloride concentrations. The buffer solutions Krebs bicarbonate (126 mM NaCl, 0.38 mM KH_2_PO_4_, 2.13 mM K_2_PO_4_, 1 mM MgCl_2_, 1 mM CaCl_2_, 24 mM NaHCO_3_, 10 mM glucose), zero-chloride buffer (116.2M sodium gluconate, 2.4 mM KH_2_PO_4_, 1.24 mM K_2_HPO_4_, 1 mM MgSO_4_, 1 mM calcium gluconate, 25 mM NaHCO_3_, 10 mM glucose), and zero-bicarbonate buffer (145 mM NaCl, 3.3 mM K_2_HPO_4_, 10 mM HEPES, 1.2 mM MgCl_2_, 1.2 mM CaCl_2_, 10 mM glucose) were used. The pH was adjusted using acetic acid [9].

### 2.4. Statistical Analyses

Statistical analyses were conducted in GraphPad (Prism 9.4.1). D’Agostino and Pearson normality tests were used to determine the normal distribution of the FLIPR measurements. Student’s paired *t*-test was performed on data with two sets. One-way ANOVA with secondary analysis using Tukey’s multiple comparison test was used to compare responses by treatment groups. Pearson and Spearman correlations were reported. Values of *p* < 0.05 were considered to be statistically significant. 

## 3. Results

### 3.1. Fluorescence-Based Measurement of F508del-CFTR Channel Activity in Apical Membranes of Patient-Derived Nasal Cultures Reports Differential Modulator Efficacies

Primary nasal epithelial cultures, derived from 31 individuals with cystic fibrosis and homozygous for F508del, were plated and differentiated in 96-transwell culture plates as described in our previous publications [12] and in the Materials and Methods section. As shown in Figure 1, typically, 12 cultures from a single donor were differentiated simultaneously to enable simultaneous replicate studies of the multiple interventions. Figure 1 shows the FMP responses mediated by nasal cultures generated from three different individuals after exposure to vehicle (DMSO) alone or CFTR correctors for 48 h. In these studies, we compared long-term treatment with lumacaftor (L), tezacaftor (T), or elexacaftor plus tezacaftor (ET). Cultures were then exposed to the FMP dye in a low-chloride buffer—a condition that facilitates the detection of CFTR channel activity. Finally, apical membrane chloride channel activity was stimulated by a cAMP agonist—forskolin plus the potentiator ivacaftor (I). CFTR-mediated depolarization was measured as an increase in fluorescence intensity that was reversible with the addition of CFTRInh-172. 

The representative data displayed in Figure 1 show the time course for F508del-CFTR channel stimulation (activation and potentiation) and inhibition by CFTRInh-172, measured as changes in fluorescence by the FMP technique. The traces were obtained during a 35 min time interval. In this example, cultures from three donors were studied for modulator responses simultaneously, in triplicate. The magnitudes of forskolin–ivacaftor-activated depolarization and CFTRinh-172-inhibitable responses were colour-coded with respect to their relative magnitudes. On the bottom panel, the dots show the peak fluorescence detected across a well scan for each individual transwell. In Appendix A, we show that the kinetics of the responses are comparable to those measured using the SpectraMax i3X plate reader, although fewer data points were captured using the SpectraMax i3X. 

The data for all of the donors who were homozygous for F508del (*n* = 31) are shown in Figure 2. Each dot represents the mean of the two most similar values of technical replicates for each donor studied. Each of the donors was homozygous for F508del, suggesting that variability in the measured channel activity may be conferred by factors other than the CFTR genotype, including environmental and genetic factors or the impact of in vitro culture conditions. For a subset of donors (*n* = 10), we confirmed that there were significant correlations between the modulator responses measured in cultures generated from two different frozen aliquots of cells, supporting the claim that the culture conditions are consistent (Appendix A). Importantly, even with this variability amongst donors, after 48 h of correction with lumacaftor (L), tezacaftor (T), or elexacaftor plus tezacaftor (ET), the stimulation mediated by forskolin and ivacaftor (I) was statistically significant. These results show that the FMP method is capable of reporting the superior efficacy of the triple combination: ETI relative to LI and TI in patient-derived nasal epithelial cultures. 

### 3.2. FMP Measurements of Nasal Cultures Generated from People with Rare Mutations Enable Testing of Emerging Therapeutic Strategies

We were then prompted to further test the capacity of this in vitro assay to report functional responses to clinically relevant modulators by nasal cultures derived from individuals with rare CF-causing mutations. The potentiator, VX-770 or ivacaftor, is prescribed as a therapy for those people who harbour Class III mutations, including G551D and G178R. We studied FMP responses in avatars generated from five individuals with G551D or G178R on one or both alleles (Table 2). On the left-hand side of Figure 3, we show the time course for constitutive and potentiated (FSK+I) mutant channel activity, with its subsequent inhibition by CFTRInh-172 in nasal cultures differentiated at the air–liquid interface on transwell filters. In the middle panels, we show dots coloured according to the peak, potentiated apical chloride conductance for each well in the plate. The scattergram in the rightmost panel shows the variable, patient-specific responses to potentiation. On aggregate, the scattergram shows that treatment with ivacaftor caused a significant increase in apical chloride conductance mediated by G551D-CFTR, as expected. 

Figure 4 shows the results of the plate reader assay with raw data for nasal cultures from three individuals who were homozygous for W1282X. The traces on the far left show that long-term (48 h) treatment with the nonsense-mediated decay inhibitor SMG1i and the read-through agonist G418 plus correctors—elexacaftor plus tezacaftor El (ET)—enhanced W1282X-CFTR-mediated channel activation by forskolin in the presence of ivacaftor. This combination was previously shown to be partially effective in promoting the functional rescue of this nonsense mutation [13,14]. The scattergram, where each symbol represents the mean of the two most similar measurements in the plate reader assay, shows that the QUAD combination caused a statistically significant augmentation of the ivacaftor-stimulated W1282X-CFTR channel function. 

### 3.3. Correlation of Electrophysiological and Fluorescence-Based Measurements of Non-CF and CF Nasal Epithelial Cultures

Finally, we asked whether there was a correlation between in vitro responses to drugs or tool compounds measured in patient-specific nasal cultures using the FMP assay and the responses measured using the Ussing chamber apparatus. As shown in Figure 5a, we found that the responses measured in cultures generated from patients homozygous for F508del using the FMP or Ussing chamber assays were correlated according to both the Pearson (r = 0.65, *p* < 0.0001) and Spearman (ρ = 0.67, *p* < 0.0001) correlation values. The numbers of patient-specific cultures harbouring G551D or G178R were fewer (*n* = 5), and the responses to the vehicle or ivacaftor exhibited a correlation only when evaluated using the Pearson test, but not the Spearman test (Figure 5b). Finally, no correlation was found between the measurements obtained in patient-specific cultures homozygous for W1282X (*n* = 3, Figure 5c). As the figure makes clear, the responses to the combination of small molecules (ETI+SMG1i and G418) were modest in the Ussing chamber, as previously documented [13,14]. On the other hand, the responses to the same combination as measured in the FMP assay were relatively robust in the three patient-specific cultures.

## 4. Discussion

Here, we show the in vitro responses to CFTR modulators measured in primary nasal epithelial cultures derived from 39 individuals representing different CFTR genotypes. We found that the fluorescence membrane potential (FMP)-based assay recapitulated the patient-specific responses to CFTR modulators measured in the Ussing chamber for matching cultures harbouring F508del. These findings support the use of this method as a complementary approach for theratyping CFTR modulator responses in primary nasal epithelial cultures.

It is important to note that the extent of rescue was variable amongst nasal cultures generated from different individuals who were homozygous for F508del. Previous studies have reported such inter-donor variation in in vitro studies [5,15,16], and the basis for this variability remains unknown. All of the cultures were generated using an identical differentiation protocol, and the CFTR genotype of F508del/F508del for each person in the cohort was identified in clinical genetics laboratories. Therefore, we speculate that the inter-donor variation could have been due to donor-specific variations in the propensity for differentiation and/or the unique genetic background of non-CFTR genes specific to each individual. Alternatively, it is possible that the inter-donor variation could be conferred by additional variants in the *CFTR* gene. Long-read sequencing is underway as part of a large study to address this question.

As shown in Figure 1, Figure 2, Figure 3 and Figure 4, the FMP assay was effective in reporting significant improvements in mutant CFTR channel function in the apical membranes of patient-derived nasal cultures in response to modulators approved as therapies for people harbouring F508del. The FMP assay reported significant responses to the laboratory treatments that modeled Orkambi (L), Symdeko (T), and Trikafta (ETI) drug treatment. Furthermore, the superior response to the modulators in Trikafta was also detected using the FMP assay in patient-derived nasal epithelial cultures. As expected, nasal tissue cultures generated from people with CF and harbouring at least one copy of G551D showed significant responses to ivacaftor in the FMP assay. Although there are no approved therapies for people harbouring W1282X on both alleles, the combination of investigational compounds previously shown to induce functional rescue of this nonsense mutation in Ussing chambers [13,14] also induced a significant functional rescue in the FMP assay. Together, these findings support the application of the FMP method to theratyping efforts, aimed at providing a guide to potential therapies or therapeutic strategies for people harbouring rare mutations.

In comparing the functional rescue of patient-derived cultures homozygous for W1282X using the two assays, we observed an interesting trend to be investigated in future studies (see Figure 4 and Figure 5). Namely, all three of the cultures derived from patients who were homozygous for W1282X showed a positive response to the combination of CFTR modulators, the nonsense-mediated decay inhibitor SMG1i, and the read-through agent G418 in the FMP assay. One possible explanation for this discrepancy may relate to the impact of the paracellular pathway on Ieq measurements in the Ussing apparatus. The leakiness of the paracellular path will reduce the amplitude of transepithelial currents measured in the Ussing chamber, and, the loss of the PDZ-binding motif in CFTR and interacting scaffolding proteins in cultures harbouring the W1282X mutation may impact this pathway [17]. On the other hand, the FMP assay reports the conduction of chloride ions across the apical membrane, independent of the paracellular pathway [12]. The intriguing observation that the FMP assay is sensitive to the functional rescue of W1282X supports its value in future therapy discovery programs that target this and other nonsense mutations.

## 5. Conclusions

This study supports previous pilot studies and shows that a mid-throughput plate reader assay of fluorescent membrane potential (FMP) recapitulates drug responses by F508del-CFTR that were previously measured in donor-specific nasal cultures using the Ussing chamber method. This convenient and accessible FMP method has the potential to facilitate inter-institutional theratyping studies as well as therapy discovery efforts targeting W1282X-CFTR.

## Figures and Tables

**Figure 1 cells-12-01174-f001:**
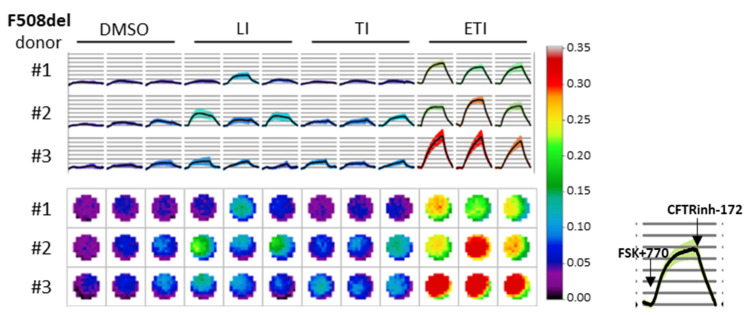
FMP plate reader studies of responses to forskolin plus ivacaftor (I) on nasal cultures from 3 donors possessing F508del on both alleles: The traces on top show reproducible responses to forskolin plus ivacaftor after long-term treatment with DMSO, L = lumacaftor, T = tezacaftor, or ET = elexacaftor plus tezacaftor. The insert to the far right shows the point in the trace when the stimulatory intervention (Fsk+I) was added and the point at which the inhibitory intervention (CFTRInh-172) was added. The total time represented by the trace = 35 min. The coloured regions in the traces are +/−1 SD of the mean at each time point. The mean is indicated as the black tracing. Traces are colour-coded according to the magnitude of response. The dots in the bottom panel represent a different visualization, with each dot showing a well scan of the FMP response across each culture surface, coloured according to magnitude. The colour scale bar indicates the percentage change in fluorescence measured using the FMP method.

**Figure 2 cells-12-01174-f002:**
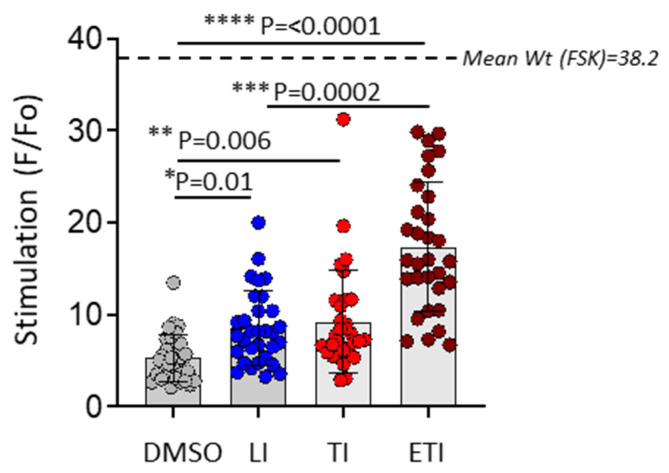
Summary of modulator responses measured in the plate reader assay (Figure 1) for 31 individuals, homozygous for F508del. Each dot represents data from cultures generated from a different individual. These cultures were treated for 48 h with the vehicle control (DMSO) and then acutely stimulated with forskolin. The cultures were also treated with correctors—lumacaftor (L, blue), tezacaftor (T, light red), or elexacaftor plus tezacaftor (ET, dark red) for 48 h. The mean forskolin response measured in non-CF cultures (*n* = 9) is shown as a stippled line for comparison. Treatment with correctors (L, T, and ET) statistically enhanced FMP responses to forskolin plus ivacaftor, as determined by one-way ANOVA with multiple comparisons test.

**Figure 3 cells-12-01174-f003:**
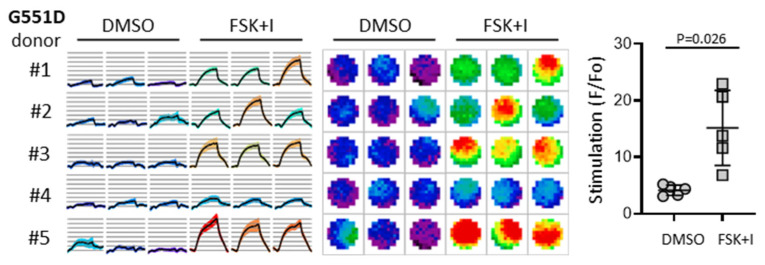
FMP plate reader studies of responses to forskolin plus ivacaftor on nasal cultures from 5 donors possessing G551D on one or both alleles. The traces on the **left** show reproducible responses to forskolin plus ivacaftor (I). The coloured regions in the traces are +/−1 SD of the mean at each time point. The mean is indicated as the black tracing. The traces are colour-coded according to the magnitude of the response. The dots in the left-middle panel represent a different visualization, with each dot showing a well scan of FMP, coloured according to magnitude. On the **right**, the mean FMP responses for the five donors are shown. There was significant augmentation of the functional expression following acute treatment with FSK+I, as determined using Student’s paired “*t*” test.

**Figure 4 cells-12-01174-f004:**
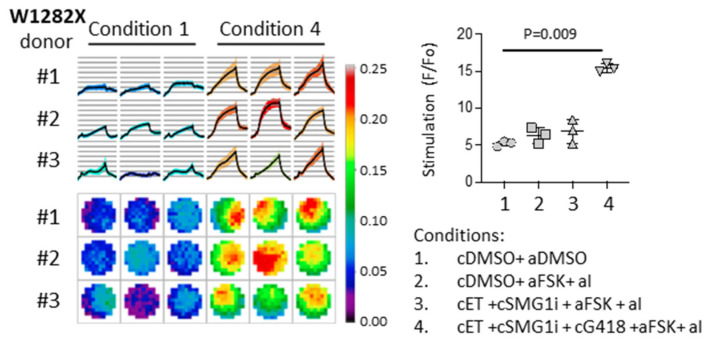
FMP plate reader studies of investigational compound responses on nasal cultures from 3 donors, each homozygous for W1282X. The traces on the (**left** (**top**)) show reproducible responses to forskolin plus ivacaftor (I) after pre-treatment with ET+SMG1i and G418. The coloured regions in the traces are +/−1 SD of the mean at each time point. The mean is indicated as the black tracing. The traces are colour-coded according to the magnitude of the response. The dots in the (**lower-left**) panel represent a different visualization, with each dot showing the peak FMP response in each culture, coloured according to magnitude. On the (**right**), the mean FMP responses for the three donors are shown. There was a significant augmentation of the functional expression following treatment with condition 4, as determined using Student’s paired “*t*” test.

**Figure 5 cells-12-01174-f005:**
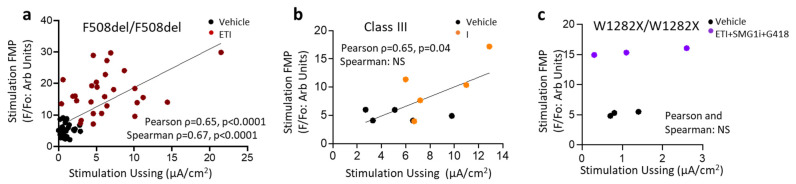
FMP correlation between donor-specific responses (i.e., stimulation) in FMP-based and Ussing chamber studies on primary nasal epithelial cultures of different genotypes: (**a**) FMP vs. Ussing responses in cultures from donors homozygous for F508del (*n* = 31). The black symbols correspond to measurements obtained in the presence of the vehicle control. The dark red symbols represent measurements obtained in the presence of ETI. (**b**) FMP vs. Ussing responses in cultures from donors harbouring G551D or G178R (*n* = 5). The black symbols denote measurements obtained using the vehicle control, while orange symbols denote the acute responses to forskolin plus ivacaftor (I). (**c**) FMP vs. Ussing responses in cultures from donors harbouring W1282X (*n* = 3). Black symbols indicate measurements in the presence of DMSO, while purple symbols represent treatment with SMG1i, G418, and the triple-modulator combination in ETI. The line in (**a**,**b**) represents the simple linear regression line fit for the data.

**Table 1 cells-12-01174-t001:** Chronic drug treatment conditions of HNECs.

HNEC	Chronic Treatment Condition
F508del/F508del	Condition 1—0.1% DMSO Condition 2—3 µM VX-809 (Lumacaftor—L) Condition 3—3 µM VX-661 (Tezacaftor—T) Condition 4—3 µM VX-661 + 3 µM VX 445 (Tezacaftor + Elexacaftor—ET)
Class III	None
W1282X/W1282X	Condition 1 and 2—0.1% DMSO Condition 3—3 µM VX-661 + 3 µM VX-445 + 0.5 µM SMG1i Condition 4—3 µM VX-661 + 3 µM VX-445 + 0.5 µM SMG1i + 200 µg/mL G418

**Table 2 cells-12-01174-t002:** CFTR genotypes of individuals who are heterozygous for Class III mutations.

Donor Number	Mutation
1–4	G551D/F508del
5	G178R/F508del

## Data Availability

The data presented in the current study can be made available upon request to the corresponding author.

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
