# Peer review of "Correlation of Electrophysiological and Fluorescence-Based Measurements of Modulator Efficacy in Nasal Epithelial Cultures Derived from People with Cystic Fibrosis"

_cells, 2023, doi:10.3390/cells12081174_

Round 1

Reviewer 1 Report

In their manuscript Gunawardena and colleagues reports the comparison between electrophysiological and fluorescence-based measurements performed on nasal epithelial cultures derived from CF subjects with different genotypes as well as from non-CF controls. Functional evaluation of CFTR activity and rescue by modulator combinations is usually performed by electrophysiological techniques such as the Ussing chamber. However, as stated by the authors, short-circuit current measurements are time consuming, require specialized equipment and expertise. In a previous work the same group reported the development of a novel fluorescence-based, multi-transwell method for assaying regulated apical chloride conductance (Fl-ACC). Such an assay would enable theratyping in patient-derived nasal cultures in any laboratory possessing a fluorescence plate reader.

In this manuscript, the authors provide experimental data to demonstrate the correlation between Ussing chamber measurements and fluorescence plate reader-based measurements of CFTR mediated apical conductance in matching, fully-differentiated nasal cultures derived from CF patients. The authors found a strong correlation between CFTR channel activity measured by Ussing and the Fl-ACC assay, suggesting its potential use in laboratory settings thereby allowing for inter-institutional replication and validation of theratyping.

Comments:

- Line 103: Please specify for how many days the epithelia were differentiated in PneumaCult ALI.

-Line 106: Please specify the type and volume of solution (for the chronic drug treatment) that was added apically. Is there a reason to treat the epithelia apically instead of basolaterally when testing CFTR modulators (for the chronic drug treatment)?

-Fig. 2: Among the F508del cohort, the extent of rescue is markedly different in different subjects. Please specify if the genotypes were confirmed for example for those very poorly responsive to drug treatment.

-Fig. 5: please prepare multiple graphs to show non only the overall correlation, but also the correlation for Ussing vs Fl-ACC assay for each genotype analyzed to highlight possible differences based on the genotype.

-What is the reproducibility of results when you perform the Fl-ACC assay on cells deriving from different frozen aliquots (of the same donor)? 

In my personal view, I would tone down the statement (suggestion?) that the Fl-ACC assay will enable theratyping in patient-derived nasal cultures in any laboratory possessing a fluorescence plate reader. While I understand the need to theratype CF people on a larger scale to promote their access to drugs, there is still the need to gain further knowledge on the mechanism of action of CFTR mutations and their response to modulators. It is probably too early to transform theratyping in a sort of “blood exam” that can be performed by anyone regardless their scientific background. 

On the other hand, I would support the use of Fl-ACC assay for drug discovery projects or to evaluate modulators efficacy (under particular condition) on a large number of subjects. As an example, it could be used to investigate the effect of acute vs chronic ivacaftor (i.e. ET vs ETI) on the F508del cohort to highlight possible interindividual differences.

Author Response

Comments to Editor and Responses to Reviewer #1

Anna Zhao

Section Managing Editor

Cells

Re: Manuscript ID: cells-2291722 Type of manuscript: Article Title: Correlation of electrophysiological and fluorescence-based measurements of modulator efficacy in nasal epithelial cultures derived from people with Cystic Fibrosis.

April 2, 2023.

Dear Anna Zhao,

Thank you for the opportunity to revise and resubmit our manuscript. We addressed all of the comments and concerns raised by our reviewers.

In our revisions, we added new experimental data (Supplementary figure 1 and 2). Supplementary figure 1 addresses a request raised by Reviewer 1, where we show the correlation in phenotypic data obtained for two different frozen aliquots from the same patient. Supplementary figure 2 addresses a request by Reviewer 2 where we show that a standard plate reader is capable of running the same fluorescence based assay as we described for the FliPR Tetra instrument. We made extensive revisions to all sections of the text and replaced figure 5  in order to address the remaining concerns raised. We uploaded a blue type manuscript where the revised text is blue to help in identifying the changes made.

Please find a point-by-point response to all of the concerns/comments raised by our reviewers in the paragraphs below.  We hope that we have satisfied their concerns and look forward to hearing your response.

Respectfully yours, Christine Bear, Ph.D.

Senior Scientist, Program in Molecular Medicine

Research Institute, Hospital for Sick Children

Toronto, CAN

bear@sickkids.ca

416-543-1067

Reviewer 1-

Comments and Responses:

Comment- Line 103: Please specify for how many days the epithelia were differentiated in PneumaCult ALI.

Response- The epithelial cultures harbouring dF508, G551D, G178R were differentiated for 14 days after transfer to transwell filters. The cultures harbouring the nonsense mutation, W1282X were differentiated for 28 days as the longer time was required to acquire the minimum transepithelial resistance of 200 Ohm cm2. The manuscript was revised to include this information (lines 96-99 in revised manuscript).

Comment - Line 106: Please specify the type and volume of solution (for the chronic drug treatment) that was added apically. Is there a reason to treat the epithelia apically instead of basolaterally when testing CFTR modulators (for the chronic drug treatment)?

Response- The modulators were added to the media bathing the basolateral surface. Care was taken to ensure that DMSO volumes were 0.1% of the total bath volume. Please see text in line 110 of the revised manuscript.  

Comment- Fig. 2: Among the F508del cohort, the extent of rescue is markedly different in different subjects. Please specify if the genotypes were confirmed for example for those very poorly responsive to drug treatment.

Response- The extent of rescue is variable amongst nasal cultures generated from people, homozygous for F508del. Our group is not the first to find this degree of inter-donor variation in in-vitro studies (Pranke et al. 2019, Matthes et al. 2018) and there is heterogeneity in clinical response size to Trikafta (Middleton et al. 2019).  The basis for this inter-donor variability remains unknown. With respect to the cultures themselves, we ensured that the culture conditions were standardized. As quality control measures, we ensured that the baseline electrical resistances exhibited by cultures were not less than 200 Ohm*cm2

The CFTR genotype of F508del/F508del for each person in the cohort was determined in clinical genetics laboratories. We speculate that the inter-donor variation is due to the unique genetic background of non-CFTR genes specific to each individual. However, we acknowledge that it is possible that the inter-donor variation could be conferred by additional variants in the CFTR gene. Long read sequencing is underway as part of a larger study to address this question. 

The authors have revised the manuscript to include these points. Please see lines: 307-317.

Comment- Fig. 5: please prepare multiple graphs to show non only the overall correlation, but also the correlation for Ussing vs Fl-ACC assay for each genotype analyzed to highlight possible differences based on the genotype.

As suggested by our reviewer, we revised figure 5 to show the correlation for Ussing vs FLIPR for each, individual genotypes: F508del, G551D and W1282X.  The patient specific, in-vitro responses to vehicle or Trikafta (ETI), in cultures harbouring dF508, showed a correlation between the measurements obtained using the FMP or Ussing techniques. This correlation, (Spearman=0.7), supports the use of the FMP method for ranking donor-specific responses to Trikafta amongst cultures generated from people harbouring dF508. Because of the low number of cultures from donors harbouring Class III mutations or homozygous for W1282X, the correlations are weak or insignificant, respectively.  Together with the graphs shown in Figure 3 and 4, we suggest that the FMP assay may be used to identify promising interventions for these rare mutations, rather than being used to report patient-specific responses. Please see lines 329-331 of the revised manuscript.

Comment- What is the reproducibility of results when you perform the Fl-ACC assay on cells deriving from different frozen aliquots (of the same donor)?  

We include new studies where different frozen aliquots from 10 donors (homozygous for dF508), were studied using the Fl-ACC assay. These data are included as a Supplementary figure, Suppl.Fig.2.   We show that there was a significant correlation between the drug responses measured using Fl-ACC for different aliquots from the same donors. Please see lines 206-209 of the revised text.

Comment- In my personal view, I would tone down the statement (suggestion?) that the Fl-ACC assay will enable theratyping in patient-derived nasal cultures in any laboratory possessing a fluorescence plate reader. While I understand the need to theratype CF people on a larger scale to promote their access to drugs, there is still the need to gain further knowledge on the mechanism of action of CFTR mutations and their response to modulators. It is probably too early to transform theratyping in a sort of “blood exam” that can be performed by anyone regardless their scientific background. 

Response- We agree that critical interpretation of drug responses in patient-specific tissues is required prior to any clinical decision making.

Reviewer 2 Report

In this manuscript, Gunwardena and colleagues apply a previously reported cellular assay for the quantification of CFTR-dependent anionic transport (Fl-ACC, based on the FLIPR Membrane Potential fluorescent dye) to primary nasal epithelial cultures with multiple CFTR genotypes of different classes (wt, F508del, G551D, G178R, W1282X) to examine the functional rescue of clinically relevant CFTR mutants by clinical and investigational molecules, including CFTR modulators. The authors observed the expected genotype-dependent functional rescue by SMG1i, G418 (W1282X, class I mutant), CFTR folding correctors lumacaftor, tezacaftor and elexacaftor (F508del, class II) and CFTR potentiator ivacaftor (G551D, G178R, class III). Finally, the authors correlated the Fl-ACC results with Ussing chamber measurements of CFTR transcellular ionic transport – one of the standard techniques for assessing CFTR activity – on the same samples and observed a significant correlation between the two, proposing Fl-ACC as an additional theratyping and drug discovery assay in the Cystic Fibrosis field. The work is relevant and has been well designed, but it is my opinion that the report still needs improvements before deserving publication.

Major points:

1.       The authors claim that, being a plate reader-based assay, Fl-ADD can be adopted in all labs which already have a plate reader in a straightforward manner. Moreover, the plate reader technology is presented in opposition to the Ussing chamber technique (claimed to require specialized equipment and expertise), meaning that Fl-ACC can be a way to overcome some of these hurdles. I strongly disagree with these claims. In fact, the plate reader used by the authors is by no means a standard plate reader. Instead, the Molecular Devices Flipr Tetra is actually a high content screening microscope with liquid handling capabilities. Most researchers associate plate reader-based assays with the acquisition of spectroscopic data, in the form of some sort of data trace. Instead, Fl-ACC is a timelapse image-based assay, which would no doubt need skills beyond those found in most research labs, especially for image and data analysis. The authors must make a fair description of the nature of the assay (image-based, not spectroscopy-based; microscope, not plate reader; non-trivial analysis, etc), include missing methodological information (what objective lens and detector were used, any relevant liquid handling-related information) and provide the image/data analysis algorithm.

2.       The references must be re-checked. Importantly, Ref 3 is not the original description of the FIS assay [line 53] and Ref 9 is not the proof-of-concept study performed by this manuscript’s authors [line 69]. Additionally, the manuscript fails to cite previous works using FLIPR or microscopy to assess CFTR activity.

3.       Table 1: Why are some negative controls performed in the absence of any chronic treatment and other performed under 0.1% DMSO?

4.       Lines 121-127. These sentences need re-writing. There is a typo: “output therefore”. The data is not actually a single TIFF image, but rather a collection of images (i.e. a timelapse). Please check the pixel size (512x341 is a rather unusual size). What is the time resolution? What is the “x” second interval between scans? The normalization strategy is not clear: shouldn’t the normalization occur in the entire image, rather than on individual pixels? Most importantly: how was image segmentation performed? Was there any cell quality control in place? Were the raw fluorescence values summarized in any way (mean fluorescence by image?). Please provide the image analysis algorithm.

5.       Statistical analysis: which ANOVA post-hoc was applied? Mention the accepted significance level in the methods section. Which data normality test was performed? The Spearman correlation coefficient is usually abbreviated by the Greek letter rho, not the lowercase “r”. Please use the rigorous terminology “Student’s t-test” instead of the colloquial “’t’ test”. Paired tests can only be performed when assessing physically the same entities under different experimental conditions. Paired tests do not seem to be adequate for the data presented in this manuscript (one well is only assayed once, different wells are treated with different compounds). Please redo the statistical tests using unpaired models.

6.       Figure 1: how do the authors explain a larger response to Tezacaftor than to Lumacaftor? Lumacaftor is currently an “orphan drug” because Tezacaftor is a much more effective analog.

7.       How do the authors explain the fact that some kinetic traces look different for the same genotype and compound treatment? Some have the typical smooth activation and deactivation profile, others have almost rectilinear activation and deactivation regions, and others how biphasic deactivation curves.

8.       The ability of SMG1i and G418 to rescue W1282X is known, unlike what the authors seem to claim. Please clarify and, if relevant, cite relevant publications.

Minor points:

1.       For the sake of clarity, I suggest that the authors convey the notion that not even the best in vitro data fully correlates with clinical outcomes. By other words, not even the well-established Ussing chamber measurements (implicitly, the baseline for the correlation analysis) nor Fl-ACC fully translate the clinical benefit for people with CF treated with CFTR modulators.

2.       There are some inconsistencies, which need uniformization. For example, one can find “F508del” and “dF508”; “VX-809” and “VX 809”; “Lumacaftor” and “VX-809” are used interchangeably without a proper presentation of both designations;

3.       The authors fail to mention which Flipr dye was used, as well as a catalog reference.

4.       The chronic treatment duration is not clear. In the methods section a 48h period is mentioned, but elsewhere (e.g. Fig 2) a 24h period is mentioned instead. Please explain/correct.

5.       Line 118: fluorescence excitation and emission usually occur as wavelength bands (not single wavelengths). Please correct.

6.       In figures 1, 3 and 4 it may not be clear at which time point VX-770, Forskolin and CFTRinh172 are added. Please indicate.

7.       Not all data in figures 1, 3 and 4 is clearly presented. What are the purple/blue/orange/red “error bars” on the kinetic tracings? What does the color scale represent? The “dots” are not actually dots, but rather well heatmaps. Was there any pixel binning involved in the generation of the heatmaps? How do the authors explain the non-homogeneous color distribution in the wells (is there any problem with cell density, liquid dispensing or image acquisition?). Please explain.

8.       The legend of Figure 1 does not seem to match the layout of the figure.

9.       Line 174: how many measurements were performed on each donor, before selecting the 2 most similar? Why was this necessary? Please define what “avatars” are.

10.    Line 181: This is actually a “triple” combination. The assay is performed in the presence of VX-770 as well.

11.    Figure 5: the meaning of the colored symbols would be better presented as a figure, rather than as a figure legend. Also, the legend fails to mention the cellular model and the number of samples in each genotype.

12.    It is not clear what the “discrepancy” mention in line 278 is. Were the authors expected to observe a rescue extent to 25% of wt-CFTR function? If so, please provide a reference.

13.    There are some typos: “throughput” repetition [line 57], “coloured coded” [line 169], “cDMSO” and “aFsk” [Fig 4], “G418X” [line 276].

Author Response

Reviewer 2

Major points:

Comment- The authors must make a fair description of the nature of the assay (image-based, not spectroscopy-based; microscope, not plate reader; non-trivial analysis, etc), include missing methodological information (what objective lens and detector were used, any relevant liquid handling-related information) and provide the image/data analysis algorithm.

Response-

We acknowledge the need to provide a better description of the Fl-ACC assay and made comprehensive changes to the methods section of the revised manuscript (see lines 121-156). 

The original FL-ACC assay of patient-derived nasal cultures as described by Ahmadi et al. (NPJ-Genomics Med.) was developed using a standard plate reader. Currently, we conduct drug testing using Fl-ACC on nasal cultures using either a SpectraMax i3X or the FLIPR Tetra.

We agree with the reviewer's comment that the FLIPR Tetra instrument is not a standard plate reader. We revised the text of the manuscript to include more detail regarding the Tetra instrument. For example, the TETRA uses an electron-multiplying CCDs (EMCCDs) sensor to enhance sensitivity of fluorescence imaging.  In addition, fluid transfer is carried out as a single aspirate-single dispense with 5ul of drug being added during each drug addition.

In this paper, we used the images from the FLIPR Tetra to provide a compelling visualization of the Fl-ACC assay. However, the principles of the Fl-ACC assays are similar regardless of the instrument used for detection. In the experiments using the Tetra, the time course for CFTR channel activation and inhibition were recorded as multiframe images, with each pixel in a well scan was turned into an individual data trace.  The analysis performed on the FLIPR Tetra-derived data was similar to the analysis for data obtained using the standard SpectraMax i3X spectrometer.   

Prompted by our reviewer, we added a new supplementary figure (Suppl. Fig. 1) to our revision.  This figure contains data from Fl-ACC assay obtained on the SpectraMax i3X plate reader.  This figure shows that similar kinetics were obtained for stimulation and inhibition using the standard plate reader as for the FLIPR Tetra.  Hence, it is not necessary to use the FLIPR Tetra to measure drug responses in patient-specific nasal cultures using the Fl-ACC assay.

Comment- The references must be re-checked. Importantly, Ref 3 is not the original description of the FIS assay [line 53] and Ref 9 is not the proof-of-concept study performed by this manuscript’s authors [line 69]. Additionally, the manuscript fails to cite previous works using FLIPR or microscopy to assess CFTR activity.

Response- We added a more comprehensive list of references for the FIS assay and fluorescence-based assays of CFTR in our revision.

Comment- Table 1: Why are some negative controls performed in the absence of any chronic treatment and other performed under 0.1% DMSO?

Response- The dF508/dF508 or W1282X/W1282X mutants were studied with chronic treatment conditions, during which 0.1% DMSO was used as a negative control of the chronic treatment conditions. However, since the Class III mutants and the healthy individuals were not subjected to any chronic treatment condition no 0.1% DMSO was added in for the treatment conditions.

Comment- Lines 121-127. These sentences need re-writing. There is a typo: “output therefore”. The data is not actually a single TIFF image, but rather a collection of images (i.e. a timelapse). Please check the pixel size (512x341 is a rather unusual size). What is the time resolution? What is the “x” second interval between scans? The normalization strategy is not clear: shouldn’t the normalization occur in the entire image, rather than on individual pixels? Most importantly: how was image segmentation performed? Was there any cell quality control in place? Were the raw fluorescence values summarized in any way (mean fluorescence by image?). Please provide the image analysis algorithm.

Response- The paragraph has been rewritten.

“The experimental raw output of the FLIPR Tetra instrument was a single, multiframe TIFF file with 105 image frames. The resolution of each frame was 512 by 341 pixels. The maximal resolution of the detector was 512 pixels and because the physical plate dimension was 127.76 by 85.47 mm that resulted in a 0.25 by 0.25 mm  pixel size. Out of the 105 frames, 5 were dedicated to measure baseline (30 s read intervals), 70 to measure stimulation (15 s read intervals) and 30 to measure the inhibition (30 s read intervals) intensities. To eliminate the intrinsic fluorescence of the plate plastic, background intensity was subtracted from the entire plate. The background intensity was calculated as the maximal intensity of the first frame excluding areas where fluorescent dye loaded cells were present (wells). Along the experiment, each pixel reported independently of the fluorescent intensity change at a given location (trace). These traces then were normalized using the last point of baseline to eliminate any absolute intensity differences due to dye loading or cell number differences, or other artifacts. Response to stimulation and inhibition were calculated as a maximal change during the corresponding experimental segment. Then the data from pixels belonging to the same transwell were averaged and transwells corresponding to the same experimental conditions were aggregated and compared”. Image segmentation was done by using the dimension specification of the 96-well transwell plates.

Comment- Statistical analysis: which ANOVA post-hoc was applied? Mention the accepted significance level in the methods section. Which data normality test was performed? The Spearman correlation coefficient is usually abbreviated by the Greek letter rho, not the lowercase “r”. Please use the rigorous terminology “Student’s t-test” instead of the colloquial “’t’ test”. Paired tests can only be performed when assessing physically the same entities under different experimental conditions. Paired tests do not seem to be adequate for the data presented in this manuscript (one well is only assayed once, different wells are treated with different compounds). Please redo the statistical tests using unpaired models.

Response- We applied the Tukey post-hoc test after one-way ANOVA and p values less than 0.05, considered significant. The normality test, D'Agostino and Pearson was performed.  Please see lines: 168-172.

The Spearman correlation coefficient has been abbreviated by the Greek letter rho in the figure 5 and the results sections. The terminology Student’s paired t-test has been used in the revised text.

With respect using the use of paired test, the authors would like to emphasize that we are considering each ID as a biological sample. Therefore, in our opinion, the paired tests (in Fig. 3) are suitable when we are comparing the response of wells containing cells derived from the same patient, i.e, the same biological sample, to different interventions.  We would also like to emphasize that for the FMP studies, each intervention is tested in triplicate, not a single well. 

Comments- Figure 1: how do the authors explain a larger response to Tezacaftor than to Lumacaftor? Lumacaftor is currently an “orphan drug” because Tezacaftor is a much more effective analog.

Response- In-vitro studies typically show that there is no difference in the response size between lumacaftor and tezacaftor (Oliver, K. et al. JCI, 2019). Similarly, we found that both compounds induce a similar effect (Figure 2).  The structural entity that mediates correction of F508del-CFTR is identical for both chemicals and both compounds interact at the same site on the CFTR  (Fiedorczuk et al. 2022).

Comments- How do the authors explain the fact that some kinetic traces look different for the same genotype and compound treatment? Some have the typical smooth activation and deactivation profile, others have almost rectilinear activation and deactivation regions, and others how biphasic deactivation curves.

Response- It is interesting that the rates of activation and deactivation are different amongst cultures generated from different donors. We don’t know the molecular basis for these differences but speculate that they reflect donor-specific variations in lipids/ protein in the apical membrane. Addressing the basis for these differences will constitute a focus for our future work.

Comments- The ability of SMG1i and G418 to rescue W1282X is known, unlike what the authors seem to claim. Please clarify and, if relevant, cite relevant publications.

Responses- We did not mean to suggest that we were the first to show that SMG1i and G418 can rescue W1282X in the presence of modulators and corrected that section of text and added recent references (de Poel, E. JCF, 2022, Keenan, MM et al. AJRCMB, 2019, Laselva, JCF, 2020).  

Minor points:

Comment- For the sake of clarity, I suggest that the authors convey the notion that not even the best in vitro data fully correlates with clinical outcomes. By other words, not even the well-established Ussing chamber measurements (implicitly, the baseline for the correlation analysis) nor Fl-ACC fully translate the clinical benefit for people with CF treated with CFTR modulators.

We agree that in-vitro measurement of patient-specific drug responses in primary tissue cultures can only be used as a guide to the potential clinical response. This statement has been included in our revisions (line 329-332). 

Comment- There are some inconsistencies, which need uniformization. For example, one can find “F508del” and “dF508”; “VX-809” and “VX 809”; “Lumacaftor” and “VX-809” are used interchangeably without a proper presentation of both designations;

Response- These inconsistencies have been addressed in the revised manuscript.

Comment- The authors fail to mention which Flipr dye was used, as well as a catalog reference.

Response- The details of the FLIPR Dye has been added in the methods section (line 122).

Comment- The chronic treatment duration is not clear. In the methods section a 48h period is mentioned, but elsewhere (e.g. Fig 2) a 24h period is mentioned instead. Please explain/correct.

Response- Correctors were added 48 hours prior to the assay.  This information has been included in our revised manuscript (line 114).

Comment- Line 118: fluorescence excitation and emission usually occur as wavelength bands (not single wavelengths). Please correct.

Response- The fluorescence wavelength ranges have been updated in the revised manuscript (Line 122).

Comment- In figures 1, 3 and 4 it may not be clear at which time point VX-770, Forskolin and CFTRinh172 are added. Please indicate.

Response- Arrows indicating the time point for addition of the acute treatment conditions have been included in a new inset for Figure 1.

Comment - Not all data in figures 1, 3 and 4 is clearly presented. What are the purple/blue/orange/red “error bars” on the kinetic tracings? What does the color scale represent? The “dots” are not actually dots, but rather well heatmaps. Was there any pixel binning involved in the generation of the heatmaps? How do the authors explain the non-homogeneous color distribution in the wells (is there any problem with cell density, liquid dispensing or image acquisition?). Please explain.

Response- The color scale represents the fluorescent intensity change due to stimulation. For instance, a value 0.20 means a 20% intensity increase of the signal compared to the fluorescent intensity of the last point of the baseline recording. The well heatmaps are colored according to the maximal (peak) intensity change at that given pixel location along the experiment (the highest nominal value of the stimulation part of the trace). Pixels were analyzed independently with no binning. The trace plot shows the average trace (traces for each pixel are averaged, in black) and the plus/minus one standard deviation for each individual time point (colored area). The magnitude of the maximal intensity change of the mean trace is represented by the color to be consistent with the color bar and the well heatmap. There are potentially multiple explanations of the non-homogeneous color distribution of the wells which again reports on the maximal response to the stimulus. Because it is a biological sample there is natural variation among the cells (slight growing conditional differences, fitness of the cells, dye loading and dye retention) which further highlights the importance of a population based assay compared to a single point reading method.

Comment - The legend of Figure 1 does not seem to match the layout of the figure.

Response- The figure legend has been changed to match the layout of the figure.

Comment- Line 174: how many measurements were performed on each donor, before selecting the 2 most similar? Why was this necessary? Please define what “avatars” are.

The in-vitro measurements were conducted in triplicate and the two most similar peak values selected for future analysis. We performed this selection based on our observation that two of the three traces were similar to one another. It will be important to determine if such selection is important as other labs reproduce this work. The term "avatar" has been used by our group and others to describe a patient-oriented tissue model and this term has been described explicitly in the text.

Comment- Line 181: This is actually a “triple” combination. The assay is performed in the presence of VX-770 as well.

Response- The correction has been made to Figure 1 to include the treatment conditions representing the VX-770 additions.

Comment- Figure 5: the meaning of the colored symbols would be better presented as a figure, rather than as a figure legend. Also, the legend fails to mention the cellular model and the number of samples in each genotype.

Response- Figure 5 has been revised along the suggestion by Reviewers #1 and 2.  However, for clarity, the meaning of the colours was indicated in the figure legend.

Comment- It is not clear what the “discrepancy” mention in line 278 is. Were the authors expected to observe a rescue extent to 25% of wt-CFTR function? If so, please provide a reference.

We removed the mention of the rescue extent to 25% of Wt-CFTR function in the revised manuscript.

Comment- There are some typos: “throughput” repetition [line 57], “coloured coded” [line 169], “cDMSO” and “aFsk” [Fig 4], “G418X” [line 276].

Response- The corrections have been made.

Reviewer 3 Report

The manuscript by Gunawardena et al. discuss the usefulness of a fluorescence-based functional assay utilizing FLIPR technology in assessing the effectiveness of CFTR modulator therapy in primary nasal epithelial cells derived from healthy individuals or patients with cystic fibrosis. The authors compare the relatively higher throughput fluorescence-based method (FMP assay) to the gold standard method of CFTR functional assay (Ussing chamber) regarding the amplitude of maximal response to modulator treatment. The assay utilizes patient-derived primary nasal epithelial cell culture that allows one to investigate patient-specific factors that are necessary to establish personalized therapy and estimate modulator response representative to individual patients. The fluorescence method discussed in this manuscript is well established and has previously been presented in numerous publications, although it was not directly compared to Ussing chamber assay in primary cell settings. The study investigates 31 hNEC derived from patients homozygous to F508del variant. The manuscript demonstrates in agreement with the literature that the combination therapy of VX-661 and VX-445 is superb compared to monotherapy with either VX-809 or VX-661. Also, the results show individual differences in the response toward modulator therapy from the same variant background, also in agreement with the literature. The same can be said for gating and nonsense variants listed in the manuscript. In summary, the manuscript demonstrates the FMP assay’s usefulness in predicting modulator response in individual patient samples with diverse genetic backgrounds. The manuscript is well-composed, and the results support the conclusions.

Critics:

·         Although the figures displaying FMP assays list 3 replicates (transwell) for each condition, the number of replicates used for statistical analysis is not stated anywhere in the text, especially for Ussing assay. Please state in the body of text or in the figure legend n=X number of replicates for each condition for statistical correctness.

·         Minor correction:

Line 119 Please correct 1μM770 to 1μM VX-770

Line 124 Please define X in the sentence of “with X number of second ….”

Line 276 Please correct typo G418X to G418.

Author Response

Reviewer 3

Critics:

Comment- Although the figures displaying FMP assays list 3 replicates (transwell) for each condition, the number of replicates used for statistical analysis is not stated anywhere in the text, especially for Ussing assay. Please state in the body of text or in the figure legend n=X number of replicates for each condition for statistical correctness.

Response- The number of replicates used for FMP assay and Ussing has now been stated in the Methods section (Lines 115, 159).

Minor correction:

Comment- Line 119 Please correct 1μM770 to 1μM VX-770

Response- Correction has been made, line 128

Comment- Line 124 Please define X in the sentence of “with X number of second ….”

Response- Correction has been made, lines 142 and 143

Comment- Line 276 Please correct typo G418X to G418.

Response- Correction has been made (line 273).
